

# A cross-sectional study evaluating insulin injection techniques and the impact of instructions from various healthcare professionals on insulin users in the southern region of Saudi Arabia

Sirajudeen Shaik Alavudeen[1], Md Sayeed Akhtar[1], Sultan Mohammed Alshahrani[1], Vigneshwaran Easwaran[1], Asif Ansari Shaik Mohammad[1], Noohu Abdulla Khan[1], Abubakr Taha Hussein[1], Salem Salman Almujri[2], Abdulrahman Saeed Alshaiban[3] and Khalid Orayj[1]

[1] Department of Clinical Pharmacy, King Khalid University, Abha, Aseer, Saudi Arabia
[2] Department of Pharmacology, King Khalid University, Abha, Aseer, Saudi Arabia
[3] Ministry of Health, King Faisal Medical City, Abha, Aseer, Saudi Arabia

Corresponding author
Sirajudeen Shaik Alavudeen,
sshaik@kku.edu.sa,
asiraj2005@gmail.com

## ABSTRACT

**Background**. It is evident that proper use of the insulin injection technique (IIT) is important for optimizing the efficacy of the therapy. Despite the readily available manufacturers' instructions, healthcare professionals (HCPs) play a major role in educating patients. This study aims to investigate the knowledge, practices, and challenges faced by insulin users regarding IIT, as well as the impact of healthcare professionals' education on it.

**Methods**. We conducted a questionnaire-based cross-sectional study using a validated online questionnaire to gather demographic and clinical data, as well as the participants' knowledge, practices, and challenges related to insulin therapy.

**Results**. The mean age of the participants was $38.25 \pm 15.58$ (mean $\pm$ SD) years, with a nearly equal distribution of genders. Thirty-six percent of the participants educated by the diabetes educators demonstrated an appropriate IIT, such as storage, priming the insulin pen (54%), skin folding (63%), injection hold time, and "use-by" date. Furthermore, the absence of diabetes education specialist training increases the likelihood of errors, potentially leading to a loss of glycemic control. Patients reported carrying insulin when traveling as one of the major challenges (27.9%), followed by timely injections (23.7%), priming (21.6%), and adjusting the insulin dose (16.8%). Forgetfulness (47.7%), traveling or altering the regular routine (15.5%), missing a meal (15.5%), and being overly busy (13.5%) were among the reasons for missing the insulin dose; all of which are easily manageable with proper education.

**Conclusion**. Consistent education and re-education are necessary for the insulin users to resolve the issues associated with suboptimal IIT. The inclusion of all stakeholders in insulin therapy, particularly the diabetes education specialists, is essential. Therefore, the Ministry of Health in Saudi Arabia should take the initiative to ensure that appropriately trained diabetes education specialists, pharmacists, nurses and other HCPs assess and follow up on patients.

## INTRODUCTION

Diabetes mellitus (DM) is a common health problem around the world, affecting significant number of individuals. Nearly 540 million people globally have diabetes, with 10.5% of them being adults between 20 and 79 years of age (*Diabetes Atlas, 2021*). Saudi Arabia ranks among the top 10 countries with high prevalence rate (24%) of DM worldwide.

DM is often associated with multiple long-term complications, including cardiovascular and cerebrovascular diseases, kidney failure, retinopathy, and neuropathy, which can affect the quality of life of the patients and lead to death (*Alavudeen et al., 2020*). The pathogenesis of DM is multifactorial. All patients with type 1 DM require insulin for life, while 20–30% of the type 2 DM patients eventually need insulin, with or without other oral medications, due to the progressive pancreatic β-cell dysfunction (*Nkonge, Nkonge & Nkonge, 2023*; *Alavudeen et al., 2020*).

For patients who require insulin to manage their diabetes, the appropriate administration of insulin is crucial. Achieving satisfactory glycemic control in insulin users depends not only on the correct titration of dosages and proper selection of insulin type but also on the proper insulin injection techniques (IIT) (*Ahmad et al., 2016*). Effective IIT is essential for optimizing therapy efficacy (*Gorska-Ciebiada, Masierek & Ciebiada, 2020*). Standard insulin injection practices mainly include proper storage of insulin, timely administration, correct IIT, rotation of injection sites, safe disposal of needles, management of hypoglycemia, and side effects (*Selvadurai et al., 2021*). Improper IIT is a common issue among insulin users. A large multinational survey involving 13,289 participants across 42 countries found that IIT was often inappropriate (*Frid et al., 2016b*). Despite significant advances in insulin delivery devices and technologies over the past decade, along with the availability of insulin technique guidelines in the public domain, there has been little improvement on how patients administer the insulin (*FIT4Diabetes, 2023*; *Gupta et al., 2024*). Additionally, a number of studies have explored the adherence to insulin therapy, treatment satisfaction, and barriers to initiating insulin therapy (*Alsaidan et al., 2023*; *Alhagawy et al., 2022*; *AlSlail & Akil, 2021*). However, research assessing the IIT of the patients in Saudi Arabia is limited.

Given the pivotal role of the healthcare professionals (HCPs) in enhancing IIT (*Mehta, Kiruthika & Laksham, 2024*; *Gorska-Ciebiada, Masierek & Ciebiada, 2020*), the influence of various HCPs on IIT warrants further investigation. Appropriate education from HCPs can greatly enhance the understanding and implementation of proper IIT, which can ultimately lead to better glycemic control. Therefore, it is reasonable to assess the insulin injection practices and the influence of education from various HCPs among the insulin users in Saudi Arabia.

The primary objective of our study is to assess the knowledge and practices of insulin users regarding IIT. The secondary objective is to explore the influence of patient education provided by various HCPs on these techniques.

## METHODOLOGY

### Study design and sample size

A questionnaire-based cross-sectional study was conducted to evaluate the knowledge and practices of the insulin users regarding proper insulin injection. The study was conducted between February 2021 and May 2021. The sample size was calculated using Raosoft software (*Raosoft, Inc, 2004*). Based on the population (*Arshad et al., 2021*) and with a margin of error at 7%, the confidence interval at 94%, and the response distribution at 50%, the required sample size was calculated as 181.

### Participants

The participants were recruited from the southern region of Saudi Arabia after obtaining their informed consent. The study included adults aged 18 and older who have been diagnosed with type 1 or type 2 DM and are currently using insulin, with or without other therapies. Individuals not on insulin therapy and those unwilling to participate were excluded.

### Study tool

Data were collected using a newly developed instrument based on the authors' experience, the relevant previous literature, regional cultural and healthcare practices, and the unique challenges faced by the participants in the study area. This tool incorporates specific items to evaluate knowledge, practices, and challenges regarding IIT (*Ahmad et al., 2016*; *Patil et al., 2017*). A 37-item questionnaire was developed and validated using the field pretest method. A pilot study was conducted, and the data was analyzed to calculate the internal consistency (Cronbach's alpha). The estimated Cronbach's alpha coefficient was found to be 0.83.

The final questionnaire comprised five sections, each containing both open-ended and closed-ended questions. The first section included eight items to collect information on the demographic and clinical characteristics of the respondents, such as age, gender, area of residence, educational qualification, occupation, type of DM, duration of DM, and assistance in insulin administration. The second section contained 10 questions to evaluate the participants' knowledge of insulin injections, while the third section comprised 12 questions designed to assess insulin injection practice. The fourth section included two questions to explore the occurrence and frequency of hypoglycemia episodes. Finally, the last section comprised five items to assess the participants' compliance with insulin therapy and the challenges they faced.

### Data collection and analysis

A convenience sampling technique was employed to collect the data. The target participants were approached and provided with a brief explanation of the study's purpose, emphasizing
voluntary participation and confidentiality. Written consent was obtained from those who are willing to participate. Participants were then given a QR code to access the online questionnaire, through which data were collected.

Data analysis was performed using the Statistical Package of Social Sciences Software, Version 21.0 (IBM, Armonk, NY, USA). We applied descriptive statistics to the categorical variables, representing them as frequency and percentages. All the categorical variables and the frequency distribution between the groups for various responses to knowledge, practice, and the associated challenges were estimated using the Chi square test, setting the level of significance between responses at a $p$-value of less than 0.05.

### Ethical consideration

The Ethical Committee of the Scientific Research, King Khalid University, approved this study (ECM#2020-168).

## RESULTS

One hundred and ninety-six participants completed the study and after eliminating the incomplete responses, the final number of responses reached to 187. The mean age of the participants was 38.25 ± 15.58 (mean ± SD) years and a nearly equal distribution of genders. Seventy percent of the study participants were bachelor's degree holders. Most of the participants (71.1%) were from urban areas. Most of our study participants administer the insulin themselves (82.9%). There were 62.6% of patients with type 1 DM, 34.2% of patients with type 2 DM, and 3.2% had gestational diabetes. The median duration of DM was 12.21 years, and the median duration of insulin treatment was 8.7 years. One third of the patients (30.5%) had received the training for IIT, followed by nurses (23%), clinicians (15%), and pharmacists (2.7%). Meanwhile, 28.9% of the study participants did not receive any training. Nearly 65.2% of the study participants had a history of hypoglycemic episodes after their insulin injection. The details are presented in Table 1.

Table 2 analyzes the differences in knowledge regarding proper IIT among participants trained by different HCPs. A considerable proportion of participants (35.7%) received training from diabetes educators demonstrated an appropriate IIT followed by the participants who received instructions from other healthcare professionals (25.4%). The majority of the participants (67%) believed that they knew how to administer the insulin properly, and they (84%) used the in-use insulin pens for less than one month. On the other hand, the participants lacked knowledge about many aspects of an appropriate IIT. For instance, 77% of the study participants stored their insulin injections, including in-use insulin pens, in the refrigerator, and only 43% of the participants cleaned the stopper before attaching the needle to the pen. We observed a significant difference in resuspending cloudy insulin before use ($p = 0.024$) and a high frequency of poor knowledge among participants who did not receive training from the diabetes education specialist.

Moreover, 80% of the study population used insulin within 15 min before meal. Even though there are different onsets and durations of action for different insulin preparations (0–15 min before or after meal for rapid acting insulin; 30–45 min before meal for short acting insulin; 10–30 min before meal for premixed insulin; 30–45 min before meal for

Table 1 Demographic characteristics of the participants.

| Demographic characteristics | | Frequency | Percent |
|---|---|---|---|
| Age in years | 38.25 ± 15.58 (Mean ±SD) | | |
| Gender | Male | 95 | 50.8 |
| | Female | 92 | 49.2 |
| Education | Illiterate | 14 | 7.5 |
| | Student | 35 | 18.7 |
| | Bachelor Degree | 131 | 70.1 |
| | Master degree and above | 7 | 3.7 |
| Area of residence | Urban | 133 | 71.1 |
| | Rural | 54 | 28.9 |
| Insulin administration | Self | 155 | 82.9 |
| | By others | 32 | 17.1 |
| Type of diabetes | Type1 DM | 117 | 62.6 |
| | Type 2 DM | 64 | 34.2 |
| | Gestational DM | 6 | 3.2 |
| Duration of diabetes | 12.21 ± 10.18 (Mean ±SD) | | |
| Duration of insulin use | 8.72 ± 3.47(Mean ±SD) | | |
| The insulin injection technique was instructed by | Clinician/Physician | 28 | 15.0 |
| | Diabetes Educator | 57 | 30.5 |
| | Pharmacist | 5 | 2.7 |
| | Nurse | 43 | 23.0 |
| | I did not received any training | 54 | 28.9 |
| History of hypoglycemic episodes | Yes | 122 | 65.2 |
| | No | 65 | 34.8 |

premixed regular insulin; 0–15 min before meal for premixed insulin analogues; no time specifics before meal for intermediate and long acting insulins), it is generally recommended to take the insulin 15 to 30 min before meal. There was a significant difference in the correct injection angle among patients trained by various HCPs. Diabetes education specialists' training was superior in this regard ($p = 0.023$). Furthermore, it was observed that 45% of the study participants maintained the injection hold-time for less than five seconds after pushing the plunger in.

The participants who received training from the diabetes educator have a high frequency of expiry date checking habits (33%), and among them, 31% brought the insulin injection to room temperature before injection. Regarding the priming of the insulin pens, most of the participants trained by the diabetes educators (54%) were priming the device by observing the drop of insulin at the needle tip. Moreover, 65% of the study participants consistently used a new needle every time before injection. Thighs and arms are the most frequently used sites for insulin injection (42% & 30%). The participants who took instructions from the diabetes educator tend to rotate the site of injection, followed by non-healthcare professionals (31.5% & 28.3%). Sixty-three percent of the study participants

**Table 2  Differences in knowledge regarding proper IIT among participants trained by different HCPs.**

| Items | Responses | Clinician/ Physician | Diabetes educator | Pharmacist | Nurse | Self-educated | Total | % | p value |
|---|---|---|---|---|---|---|---|---|---|
| Do you think your IIT is correct? | Yes | 21 | 45 | 4 | 24 | 32 | 126 | 67 | 0.066 |
| | Not sure | 7 | 12 | 1 | 19 | 22 | 61 | 33 | |
| Where do you store your in-use insulin pen? | Refrigerator | 25 | 42 | 5 | 32 | 40 | 144 | 77 | 0.324 |
| | Room temperature | 3 | 15 | 0 | 11 | 14 | 43 | 23 | |
| Before the injection, do you clean the skin with disinfectant? | No | 7 | 11 | 0 | 12 | 20 | 50 | 27 | 0.169 |
| | Yes | 21 | 46 | 5 | 31 | 34 | 137 | 73 | |
| Before attaching the needle, do you clean the stopper with disinfectant? | No | 11 | 32 | 2 | 26 | 36 | 107 | 57 | 0.167 |
| | Yes | 17 | 25 | 3 | 17 | 18 | 80 | 43 | |
| If you use cloudy insulin, do you re-suspend the insulin before use? | No | 1 | 14 | 1 | 10 | 20 | 46 | 25 | 0.024 |
| | Yes | 27 | 43 | 4 | 33 | 34 | 141 | 75 | |
| What is the timing between insulin injections and meals? | 0 to 15 min | 23 | 44 | 5 | 36 | 42 | 150 | 80 | 0.739 |
| | 16 to 30 min | 3 | 7 | 0 | 4 | 10 | 24 | 13 | |
| | More than 30 min | 2 | 6 | 0 | 3 | 2 | 13 | 7 | |
| What is the angle for inserting the needle of an insulin pen into the skin? | 45 degree | 1 | 4 | 0 | 4 | 4 | 13 | 7 | 0.023 |
| | 60 degree | 7 | 13 | 1 | 3 | 5 | 29 | 16 | |
| | 90 degree | 16 | 33 | 3 | 24 | 21 | 97 | 52 | |
| | Don't know | 4 | 7 | 1 | 12 | 24 | 48 | 26 | |
| How long do you keep the needle under the skin after injecting insulin | Less than 5 s | 13 | 18 | 2 | 24 | 27 | 84 | 45 | 0.071 |
| | 5 to 10 s | 10 | 27 | 3 | 7 | 13 | 60 | 32 | |
| | More than 10 s | 2 | 6 | 0 | 6 | 11 | 25 | 13 | |
| | I remove the needle immediately | 3 | 6 | 0 | 6 | 3 | 18 | 10 | |
| How long do you use your prefilled insulin pen after first use? | 4-6 weeks | 25 | 47 | 4 | 36 | 45 | 157 | 84 | 0.944 |
| | Continue Until the all the insulin has been used up | 3 | 10 | 1 | 7 | 9 | 30 | 16 | |
| Total | | 28 | 57 | 5 | 43 | 54 | 187 | 100 | |

**Table 3   Differences in practice regarding proper IIT among participants trained by different HCPs.**

| Items | Responses | Clinician/ Physician | Diabetes educator | Pharmacist | Nurse | Self- educated | Total | % | *p* value |
|---|---|---|---|---|---|---|---|---|---|
| Do you check the expiry date of your insulin? | No | 4 | 8 | 0 | 6 | 10 | 28 | 15 | 0.827 |
| | Yes | 24 | 49 | 5 | 37 | 44 | 159 | 85 | |
| Do you bring insulin to room temperature before injecting it? | No | 10 | 24 | 1 | 19 | 27 | 81 | 43 | 0.593 |
| | Yes | 18 | 33 | 4 | 24 | 27 | 106 | 57 | |
| Do you use new needle for every injection? | No | 8 | 22 | 0 | 17 | 19 | 66 | 35 | 0.421 |
| | Yes | 20 | 35 | 5 | 26 | 35 | 121 | 65 | |
| Do you prime the device by observing drop of insulin at needle tip? | No | 3 | 12 | 0 | 5 | 8 | 28 | 15 | |
| | Yes | 13 | 34 | 4 | 23 | 27 | 101 | 54 | 0.365 |
| | I have no Idea about priming. | 12 | 11 | 1 | 15 | 19 | 58 | 31 | |
| Which site do you use most? | Abdomen | 3 | 20 | 1 | 10 | 8 | 42 | 22 | |
| | Thigh | 15 | 23 | 4 | 19 | 18 | 79 | 42 | 0.38 |
| | Buttocks | 1 | 1 | 0 | 4 | 3 | 9 | 5 | |
| | Arm | 9 | 13 | 0 | 10 | 25 | 57 | 30 | |
| Do you rotate injection sites? | No | 4 | 6 | 1 | 6 | 8 | 25 | 13 | 0.949 |
| | Yes | 24 | 51 | 4 | 37 | 46 | 162 | 87 | |
| Do you make a skin fold? | No | 10 | 20 | 1 | 14 | 24 | 69 | 37 | 0.666 |
| | Yes | 18 | 37 | 4 | 29 | 30 | 118 | 63 | |
| Are your injection sites inspected in each visit? | No | 17 | 39 | 3 | 33 | 36 | 128 | 68 | 0.664 |
| | Yes | 11 | 18 | 2 | 10 | 18 | 59 | 32 | |
| Do you have any swelling or lumps under the skin at your usual injection sites that have been there for some time (weeks, months or years)? | No | 16 | 38 | 4 | 29 | 29 | 116 | 62 | 0.461 |
| | Yes | 12 | 19 | 1 | 14 | 25 | 71 | 38 | |
| Does insulin ever leak out of your injection site on the skin? | Yes | 13 | 12 | 0 | 16 | 19 | 60 | 32 | 0.048 |
| | No | 15 | 45 | 5 | 27 | 35 | 127 | 68 | |
| Do you massage the site of injection after injection? | Yes | 22 | 38 | 3 | 26 | 33 | 122 | 65 | 0.534 |
| | No | 6 | 19 | 2 | 17 | 21 | 65 | 35 | |
| Do you ever inject through your clothing? | Yes | 7 | 10 | 0 | 8 | 12 | 37 | 20 | 0.711 |
| | No | 21 | 47 | 5 | 35 | 42 | 150 | 80 | |
| Do you ever miss or skip an injection? | Never | 3 | 14 | 0 | 10 | 5 | 32 | 17 | |
| | Rarely | 17 | 22 | 3 | 16 | 18 | 76 | 41 | 0.029 |
| | Sometimes | 5 | 18 | 2 | 15 | 19 | 59 | 32 | |
| | Always | 3 | 3 | 0 | 2 | 12 | 20 | 11 | |
| Total | | 28 | 57 | 5 | 43 | 54 | 187 | 100 | |

make skinfolds during their insulin injection, and those trained by the diabetes educators are performing well in this regard. Thirty-eight percent of the study participants reported lumps or swelling at the injection sites. Subjects educated by different HCPs observed less insulin leakage during the insulin injection ($p = 0.048$). Many participants (65%) massage their injection site after the injection, while only 20% take their insulin injection through clothing. We found a statistically significant difference in terms of adherence toward IIT among the participants who were educated by different HCPs ($p = 0.029$), as described in Table 3.

The hypoglycemic episodes were predominant among the patients who were not following the IIT properly. The hypoglycemic events were more common with the patients who did not check the expiry date of the insulin injection, who did not bring the insulin injection to room temperature, and who did not check the injection site regularly for lipohypertrophy ($p = 0.015$, $0.056$, and $0.032$, respectively). Additionally, there was a statistically significant difference in the development of hypoglycemic episodes between the respondents who developed swelling and lumps on the injected site and those who did not develop any swelling or lumps ($p = 0.015$). The insulin leak at the injection site was found to be one of the factors impacting the development of hypoglycemic episodes ($p = 0.054$). This is further described in Table 4.

We looked into the challenges associated with the insulin injection and the reasons for missing the dose. Carrying insulin while traveling is one of the major challenges (27.9%) reported by the patients. The other challenges include taking injections on time (23.7%), preparing the injection site and priming it (21.6%), adjusting the insulin dose (16.8%), and taking injections during busy hours (16.8%). Other challenges reported by the participants are the number of daily injections, pain associated with the injections, and the complex insulin regimen (15.3%, 13.2%, and 4.2%). Moreover, the participants reported various reasons for missing the dose, including forgetfulness (47.7%), traveling or altering the regular routine (15.5%), missing a meal (15.5%), and being too busy (13.5%). Some patients (6.7%) reported feeling embarrassed to inject in public. Further details are presented in Figs. 1 and 2.

## DISCUSSION

DM is a chronic metabolic disorder and a major health issue worldwide (*Shaik Alavudeen et al., 2019*). It is a progressive disease that can affect almost all the organ systems. Maintaining optimal glycemic control is crucial for preventing or delaying DM-related complications (*Al-Hadhrami et al., 2024*; *Alavudeen et al., 2013*). Insulin is often a key treatment option for DM, used either as a monotherapy or in combination with other therapies to achieve desired glycemic control. Although the healthcare systems attempt to minimize the medication errors, insulin remains one of the high-alert medications demanding additional attention for proper administration (*Taylor et al., 2018*).

Appropriate IIT can enhance the beneficial effects of insulin (*Patil et al., 2017*). Conversely, inappropriate IIT can result in inconsistent insulin levels, poor glycemic control, and a significant increase in DM-related complications (*Tosun et al., 2019*; *Sami et al., 2017*). A previous study found that 75% of the participants did not follow the IIT and the storage instructions recommended by the manufacturers (*Milligan, Krentz & Sinclair, 2011*). Therefore, it is essential to understand how rational the insulin users are in their insulin administration and the influence of education related to IIT provided by various HCPs (*Gorska-Ciebiada, Masierek & Ciebiada, 2020*).

The results of our study clearly indicate a significant gap between the insulin injection recommendations and current practices among the participants. The majority of patients were storing their insulin pens in the refrigerator. Considering the average temperature of

**Table 4  Hypoglycemic episodes among the participants with regard to IIT practice.**

| Items | Responses | Positive hypoglycemic episodes (Frequency) | Negative hypoglycemic episodes (Frequency) | Total | % | *p* value |
|---|---|---|---|---|---|---|
| Do you check the expiry date of your insulin? | No | 6 | 22 | 28 | 15 | 0.108 |
| | Yes | 59 | 100 | 159 | 85 | |
| Do you bring insulin to room temperature before injecting it? | No | 36 | 45 | 81 | 43 | 0.015 |
| | Yes | 29 | 77 | 106 | 57 | |
| Do you use new needle for every injection? | No | 17 | 49 | 66 | 35 | 0.046 |
| | Yes | 48 | 73 | 121 | 65 | |
| Do you prime the device by observing drop of insulin at needle tip? | No | 12 | 16 | 28 | 15 | |
| | Yes | 33 | 68 | 101 | 54 | 0.605 |
| | I have no Idea about priming. | 20 | 38 | 58 | 31 | |
| Which site do you use most? | Abdomen | 14 | 28 | 42 | 22 | |
| | Thigh | 29 | 50 | 79 | 42 | 0.850 |
| | Buttocks | 4 | 5 | 9 | 5 | |
| | Arm | 18 | 39 | 57 | 30 | |
| Do you rotate injection sites? | No | 12 | 13 | 25 | 13 | 0.135 |
| | Yes | 53 | 109 | 162 | 87 | |
| Do you make a skin fold? | No | 29 | 40 | 69 | 37 | 0.110 |
| | Yes | 36 | 82 | 118 | 63 | |
| Are your injection sites inspected in each visit? | No | 38 | 90 | 128 | 68 | 0.032 |
| | Yes | 27 | 32 | 59 | 32 | |
| Do you have any swelling or lumps under the skin at your usual injection sites that have been there for some time (weeks, months or years)? | No | 48 | 68 | 116 | 62 | 0.015 |
| | Yes | 17 | 54 | 71 | 38 | |
| Does insulin ever leak out of your injection site on the skin? | Yes | 15 | 45 | 60 | 32 | 0.054 |
| | No | 50 | 77 | 127 | 68 | |
| Do you massage the site of injection after injection? | Yes | 43 | 79 | 122 | 65 | 0.848 |
| | No | 22 | 43 | 65 | 35 | |
| Do you ever inject through your clothing? | Yes | 8 | 29 | 37 | 20 | 0.061 |
| | No | 57 | 93 | 150 | 80 | |
| Do you ever miss or skip an injection? | Never | 12 | 20 | 32 | 17 | |
| | Rarely | 28 | 48 | 76 | 41 | 0.892 |
| | Sometimes | 19 | 40 | 59 | 32 | |
| | Always | 6 | 14 | 20 | 11 | |
| Total | | 65 | 122 | 187 | 100 | |

15–26 °C in the study area (*Arshad et al., 2021*), an opened insulin pen can be stored at room temperature for six weeks (*Bahendeka et al., 2019*). It is well known that the temperature variation leads to accumulation of air in the pen, which inversely affects insulin delivery at its intended time (*Ginsberg, Parkes & Sparacino, 1994*). Therefore, diabetes patients require

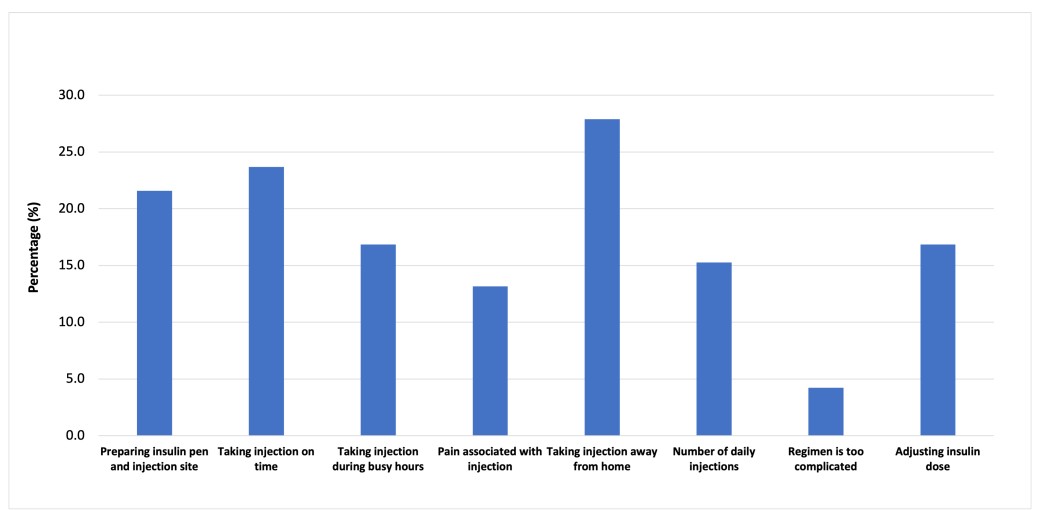

**Figure 1  Challenges faced by the participants towards insulin injection.**

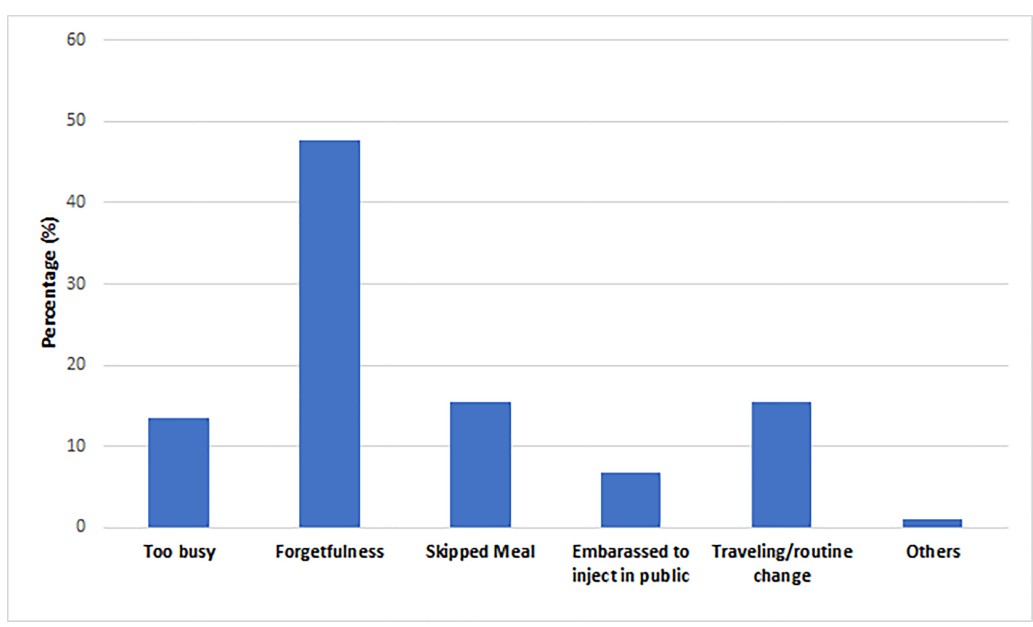

**Figure 2  Reasons for missing the dose of insulin.**

education about temperature variations and the appropriate duration of storing insulin pens to maintain the insulin potency (*American Association of Diabetes Educators, 2020*).

Moreover, nearly 35% of the participants did not check the expiry date of their insulin before injection. Additionally, 16% of the participants mentioned that they would continue to use the insulin pen until all the insulin was used up. Insulin users should be educated about the actual expiry date of their insulin, as well as the expiration of insulin after being opened. The dates can vary significantly from the expiration date printed on the insulin

pen. The expiration period for an insulin pen can differ (typically four to six weeks from the date of opening) depending on the type of insulin (*Heinemann et al., 2021*). It is advisable to educate the patients to set reminders for the "use-by" date. They should be instructed to mark their start date of insulin pen usage and set a reminder on the insulin pen's end date, because the expiration date printed on the insulin pen no longer applies once it is opened.

Guidelines recommend priming the insulin pen, resuspending cloudy insulin, and adhering to injection hold times to enhance insulin efficiency (*Mitchell, Porter & Beatty, 2012*; *Sangwan et al., 2019*). Failure to follow any of these recommendations may result in improper insulin delivery. Priming the insulin pen is a crucial step to ensure free and unobstructed flow of insulin prior to administration (*Škrha, 2022*). In our study, half of the patients reported skipping the priming step, consistent with the findings from another study (*Berard & Cameron, 2015*), which may be attributed to lack of education. Additionally, one-third of the patients reported skipping resuspending the cloudy insulin, which in turn potentially altering the clinical response and increased the insulin requirement (*Berard & Cameron, 2015*).

Holding needle in place under the skin (injection hold time) for a specified time reduces the risk of insulin leaking out of the injection site and ensures the full dose of insulin is delivered (*Sangwan et al., 2019*). Approximately one-third of the patients did not practice the injection hold time as specified by the manufacturer. Generally, the injection hold time ranges from five to 10 s for most of the insulin preparations, depending on the manufacturer. For instance, Sanofi Aventis SoloSTAR recommends a hold time of 10 s, Novo Nordisk FlexPen recommends a hold time of six seconds, and Eli Lily KwikPen or Eli Lily original disposable pen recommends a hold time of five seconds (*Mitchell, Porter & Beatty, 2012*). The variations in injection hold times may cause confusion and lead to incorrect practices if a patient switches from one manufacturer's pen to another.

Initial education provided by the HCPs is crucial for both patients and their caregivers, particularly those who are unfamiliar with priming, resuspension, and the proper hold time for insulin pens. Caregivers play a vital role in supporting patients during self-administration, making it essential for them to understand appropriate IIT thoroughly (*Sexson, Lindauer & Harvath, 2017*). The study indicates that many patients use their upper arms for insulin injection, but self-administration in this area can cause a higher risk of improper technique, leading to inconsistent insulin absorption. On the other hand, self-administration is more feasible in the abdomen and the thigh, where patients typically have good accessibility and visibility.

Patients educated by the physicians and diabetes educators demonstrate greater knowledge regarding the correct usage of insulin pens. Despite the availability of the instructional guides from the manufacturers and online instructional videos, many patients are unaware of those resources (*Mitchell, Porter & Beatty, 2012*). Our study found statistically significant differences in various knowledge and practice items, including the angle of injection, the resuspension of the cloudy insulin before injection, the injection hold time, and the incidence of insulin leakage from the injection site.

Another important aspect of appropriate insulin administration is the rotation of sites to reduce the risk of insulin-induced lipodystrophy, including lipohypertrophy and

lipoatrophy. Lipohypertrophy is characterized by a lump of fatty tissue under the skin resulting from repeated insulin injection at the same site. It is a well-known complication of insulin therapy, and patients tend to continue injecting insulin into lipohypertrophic site as they do not feel any pain. In contrast, lipoatrophy or local fat loss is less common than lipohypertrophy but can affect insulin absorption and can lead to poor glycemic control (*Gorska-Ciebiada, Masierek & Ciebiada, 2020*; *Tsadik et al., 2018*; *Kadiyala, Walton & Sathyapalan, 2014*). Therefore, insulin users should be educated on the importance of rotating injection sites.

Notably, 87% of our participants practiced injection site rotation, which aligns with findings from a similar study (*Frid et al., 2016a*; *Frid et al., 2016b*). Furthermore, we found that participants who received training from diabetes educators and nurses demonstrated a better practice in injection site rotation compared to those who were self-educated. Additionally, a significant number of the participants trained by the HCPs showed acceptable proficiency with the lifted skin fold technique compared to the self-educated patients. These findings emphasize the need for education on appropriate IIT by the HCPs, particularly diabetes care and education specialists who have specialized experience in caring for people with diabetes and related conditions (*Ryan et al., 2020*; *Down & Kirkland, 2012*).

We also have observed a high prevalence of challenges related to insulin self-administration among participants, which may hinder achieving target glycemic control. The primary issues include integrating insulin therapy into their daily routine, such as carrying insulin while traveling, taking injections on time, adjusting insulin doses, and taking insulin amid their working schedules. These challenges are aligned with the findings of a similar study conducted to identify the barriers to optimal insulin use (*Ellis, Mulnier & Forbes, 2018*). The fact that approximately 25% of participants were particularly vulnerable to omission and missed injections on time is concerning. Glycemic control tends to be poor among those who struggle to choose the correct dose and those who do not take insulin at the correct times (*Trief et al., 2016*).

Our data suggests that the patients who are not trained by a diabetes education specialist may make errors in insulin use and administration, leading to loss of glycemic control. It has been reported that the patients on insulin become more prone to erratic IIT as the duration of insulin use increases, potentially due to forgetfulness or deliberately neglecting certain steps over time. Therefore, the consistent implementation of patient education and re-education programs is essential to address the challenges associated with suboptimal IIT. This initiative should involve all stakeholders, particularly diabetes education specialists, physicians, and pharmacists, who are actively involved in patient education. Such an approach will ensure that this patient population attains the full therapeutic benefits of insulin therapy (*Frid et al., 2016a*; *Frid et al., 2016b*; *Strauss, 2014*; *Kalra et al., 2023*).

## LIMITATIONS

This study was conducted only in the southern region of Saudi Arabia, which restricts the generalizability of the results. Hence, further studies need to be carried out nationwide to explore the issues associated with insulin injection practices among the insulin users.

## CONCLUSION

Our results suggest that consistent education and re-education are essential for insulin users to address issues associated with suboptimal IIT. All stakeholders in insulin therapy should be involved, particularly the diabetes education specialists. Therefore, to ensure optimal insulin use and to attain the full therapeutic potential, healthcare authorities and the Ministry of Health in Saudi Arabia should implement initiatives that ensure patients are assessed and followed up by appropriately trained diabetes education specialists, pharmacists, nurses, and other HCPs.

### Funding

This work was supported by the Deanship of Research and Graduate Studies at King Khalid University through Large Research Project under grant number RGP2/317/45. The funders had no role in study design, data collection and analysis, decision to publish, or preparation of the manuscript.

### Grant Disclosures

The following grant information was disclosed by the authors:
The Deanship of Research and Graduate Studies at King Khalid University through Large Research Project: RGP2/317/45.

### Competing Interests

The authors declare there are no competing interests.

### Author Contributions

- Sirajudeen Shaik Alavudeen conceived and designed the experiments, performed the experiments, authored or reviewed drafts of the article, and approved the final draft.
- Md Sayeed Akhtar conceived and designed the experiments, analyzed the data, prepared figures and/or tables, and approved the final draft.
- Sultan Mohammed Alshahrani analyzed the data, prepared figures and/or tables, and approved the final draft.
- Vigneshwaran Easwaran conceived and designed the experiments, performed the experiments, authored or reviewed drafts of the article, and approved the final draft.
- Asif Ansari Shaik Mohammad performed the experiments, authored or reviewed drafts of the article, and approved the final draft.
- Noohu Abdulla Khan analyzed the data, prepared figures and/or tables, and approved the final draft.
- Abubakr Taha Hussein performed the experiments, authored or reviewed drafts of the article, and approved the final draft.
- Salem Salman Almujri analyzed the data, prepared figures and/or tables, and approved the final draft.

- Abdulrahman Saeed Alshaiban analyzed the data, prepared figures and/or tables, and approved the final draft.
- Khalid Orayj analyzed the data, prepared figures and/or tables, and approved the final draft.

## Human Ethics

The following information was supplied relating to ethical approvals (i.e., approving body and any reference numbers):

The Ethical Committee of the Scientific Research, King Khalid University, approved this study (ECM#2020-168).

## Ethics

The following information was supplied relating to ethical approvals (i.e., approving body and any reference numbers):

The Ethical Committee of the Scientific Research, King Khalid University.

## Data Availability

The raw data is available in the Supplemental File.

## Supplemental Information

Supplemental information for this article can be found online at http://dx.doi.org/10.7717/peerj.19394#supplemental-information.

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
