# Peer review of "A cross-sectional study evaluating insulin injection techniques and the impact of instructions from various healthcare professionals on insulin users in the southern region of Saudi Arabia"

_PeerJ, doi:10.7717/peerj.19394_

## Round 0.1 · original submission · Major Revisions

Dear Dr. Shaik Alavudeen,

Your manuscript entitled “A cross-sectional study evaluating insulin injection techniques and the impact of instruction from various healthcare professionals on insulin users in southern region of Saudi Arabia", which you submitted to PeerJ, has been reviewed by the editor and 3 external reviewers.

The reviewers generally support your work but have raised significant concerns that must be addressed before the manuscript can move forward. In particular, please consider clarifying the introduction, improving methodology and statistics, justifying the tool developed, and updating references.

Addressing these issues will improve your work's readability and strengthen its scientific rigor and clinical relevance. I would happily reconsider your manuscript if you undertake these substantial revisions and resubmit.

If you decide to resubmit the revised version, please summarize all the improvements made in the new version and give answers to all critical points raised in the reviewers’ report in an accompanying letter. Copy and paste each and every reviewer's comment above your response. A fluent English speaker should also check the manuscript for spelling and grammar.

Please note that resubmitting your manuscript does not guarantee eventual acceptance. Since the requested changes are major, the revised manuscript will undergo a second round of review by the same reviewers. I must emphasize that the acceptability of the revision will depend upon the resolution of the points raised by the reviewers.

Sincerely yours,

Stefano Menini

**Language Note:** The Academic Editor has identified that the English language must be improved. PeerJ can provide language editing services - please contact us at [email protected] for pricing (be sure to provide your manuscript number and title). Alternatively, you should make your own arrangements to improve the language quality and provide details in your response letter. – PeerJ Staff

·

Basic reporting

The author's choice of topic is highly significant in the management of diabetes. They have thoroughly addressed all relevant areas of interest, including those that are often overlooked.
The language used is also commendable, though minor corrections are needed.

Experimental design

The study design is appropriate; however, certain aspects of the research questions require clarification. Furthermore, the study population, as specified by the author, is limited in size and confined to the southern region, which may introduce a potential source of bias.

Validity of the findings

The study is likely to have a significant positive impact on the overall healthcare system, particularly in improving the management of diabetes within the region. By addressing key aspects in the injection techniques, the findings could enhance both clinical practices and patient outcomes. Furthermore, the recommendations put forth by the author are well-considered and reflect a deep understanding of the challenges faced in diabetes management with regard to the practice of proper injection techniques. These thoughtful suggestions, if implemented, have the potential to optimize treatment strategies and improve adherence to best practices, ultimately contributing to more effective and sustainable healthcare solutions.

Additional comments

Very informative and a good read.

·

Basic reporting

The research is good but needs some error bars to clarify the relationships and also needs accuracy in statistics.
1_ clear english
2_ sufficient literature reference
3_ need some figure (error bar)

Experimental design

good method to describe the method

Validity of the findings

Statistics in a country where there is no previous study

Additional comments

Thank you very much for the efforts made in writing this research. It needs some simple modifications regarding the statistics and clarifying the rest of the relationships.

·

Basic reporting

The introduction is somewhat limited and does not fully align with the title and aim of the study. It would benefit from further expansion to provide a clearer connection between the research focus and the broader context. Additionally, the procedure section lacks clarity, particularly in how it addresses both Type 1 and Type 2 diabetes, as these conditions differ in their treatment approaches and insulin usage. It is important to clarify how the inclusion of both types was managed in the study. Furthermore, the tool developed by the researcher requires justification. It is unclear why a new tool was necessary, given the absence of any standard instrument to measure the study's aims. Providing this rationale would strengthen the credibility of the methodolo

Experimental design

Methodology and Tool Development:
The procedure section lacks clarity, particularly regarding the inclusion of both Type 1 and Type 2 diabetes. These conditions differ in insulin usage and treatment regimens, so the approach for including both in the study needs to be clearly explained.

Validity of the findings

Additionally, the tool developed by the researcher requires further justification. It is unclear why a new instrument was needed when there may be existing, standard tools available to measure the study's aims. Providing a clear rationale for the development of a new tool would strengthen the methodology.

Additional comments

update your references.

---

## Round 0.2 · Minor Revisions

Dear Dr. Shaik Alavudeen,

The manuscript is almost ready for publication, but there is still a minor comment from Reviewer 1 that you may want to address.

If you are willing to do this, I would not need to return the manuscript to the reviewers, but it could be accepted for publication.

Sincerely yours,

Stefano Menini

·

Basic reporting

No comment

Experimental design

No comment

Validity of the findings

No comment

Additional comments

Very well written article. One thing which might require a revision would be with regard to injection site. Abdomen is the most recommended injection site, followed by thighs and upper arm. Injections over upper arm are generally administered by a caregiver, as self administration in this area may increase risk of improper techniques.

·

Basic reporting

the article structure is professional

Experimental design

The language is very good

Validity of the findings

the research is valid

·

Basic reporting

Dear Editor
Greetings

The authors did the required modifications.

Thank you

Experimental design

Research question well defined, relevant & meaningful. It is stated how research fills an identified knowledge gap.

Validity of the findings

All underlying data have been provided; they are robust, statistically sound, & controlled.

Additional comments

NIL

---

## Round 0.3 · accepted · Accept

Dear Dr. Shaik Alavudeen,

Thank you for submitting the revised version of your manuscript. I have personally reviewed the revision and confirmed that all the reviewers' comments have been adequately addressed. The quality of the manuscript has significantly improved as a result. I am pleased to inform you that your manuscript is now ready for publication in PeerJ in its current form.

Sincerely yours,

Stefano Menini